# Deep-Learning Spatial Principles from Deterministic Chemical Transport Model for Chemical Reanalysis: An Application in China for PM$_{2.5}$

Baolei Lyu[1], Ran Huang[2], Xinlu Wang[2], Weiguo Wang[3], Yongtao Hu[4]

[1]Huayun Sounding Meteorological Technology Co. Ltd., Beijing 100081, P. R. China

[2]Hangzhou AiMa Technologies, Hangzhou, Zhejiang 311121, P. R. China

[3]I.M. System Group, Environment Modeling Center, NOAA/National Centers for Environmental Prediction, College Park, Maryland 20740, United States

[4]School of Civil and Environmental Engineering, Georgia Institute of Technology, Atlanta, Georgia 30332, United States

*Correspondence to*: Baolei Lyu (baoleilv@foxmail.com), Ran Huang (ranhuang2019@163.com)

**Abstract.** Well-estimated air pollutant concentration fields are critically important to compensate the observations that are only sparsely available, especially over non-urban areas. Previous data fusion methods generally used statistical models to relate observations of target variables with proxy data and supporting variables at known stations. In this study, we developed a new data fusion paradigm by designing a deep learning model framework and workflow to learn multi-variable spatial correlations from Chemical Transport Model (CTM) simulations, before using it to estimate PM$_{2.5}$ reanalysis fields from station observations. The model was composed of two modules as an explainable PointConv operation to pre-process isolated observations and a regression grid-to-grid network to build correlations among multiple variables. The model was trained with only CTM simulations and supporting geographical covariates. The trained model was evaluated in two aspects of 1) reproducing raw PM$_{2.5}$ CTM simulations and 2) generating reanalysis/fused PM$_{2.5}$ fields. First, the model was able to well reproduce the CTM simulations on full domain from sampled CTM data items at sparse locations with an average R$^2$=0.94 and RMSE=4.85 μg/m$^3$. Second, the fused PM$_{2.5}$ fields estimated from observations achieved a good performance with R$^2$=0.77 (RMSE=14.29 μg/m$^3$) and R$^2$=0.84 (RMSE=12.96 μg/m$^3$) respectively evaluated at the stringent city-level and station-level. The generated reanalysis PM$_{2.5}$ fields have complete spatial coverage within the modelling domain. One significant benefit of the fusion framework is that the model training does not rely on observations, which can be used to predict PM$_{2.5}$ fields in newly-setup observation networks such as those using portable sensors. Meanwhile in the prediction procedure, only station observations are used along with supporting covariates. The fusion model has high computing efficiency (<1s/day) due to acceleration using GPU. As an alternative to generate chemical reanalysis fields, the method can be readily implemented at near real-time and be universally applied for other simulated variables that with measurements available.

## 1 Introduction

Pollutant concentration fields with high accuracies are important for evaluating health effects, climate changes and agricultural studies (Bell et al., 2007; Donkelaar et al., 2015; Gao et al., 2017). Long-term and reliable air quality dataset could also be used to assess pollutant emission control measures (Wang et al., 2010). Data fusion method has been widely used to obtain accurate and spatially complete datasets, such as fusing air quality model simulations and station air pollutant observations to estimate fine-scale air pollutant concentration fields (Berrocal et al., 2012; Rundel et al., 2015).

Most previous studies similarly used a general paradigm to develop well-estimated air pollutant concentration fields. In this paradigm, statistical models were trained to describe non-linear relationships between observations and proxy data and

supporting variables at the locations of observation sites (Berrocal et al., 2012; Lyu et al., 2019; Chu et al., 2016). The widely used proxy data for PM$_{2.5}$ concentrations include Aerosol Optical Depth (Lv et al., 2016) and chemical transport model (CTM) simulations (Lyu et al., 2019). Popular statistical models include linear mixed effect model (Hao et al., 2016), machine learning models of random forest (Brokamp et al., 2018; Huang et al., 2021), deep neural networks (Qi et al., 2018), and ensembled models (Xiao et al., 2018). The fitted models were then used to predict concentration field of target variables in the whole area directly or through spatial spreading techniques such as Bayesian estimation (Xu et al., 2016), partial linear regression (Wang et al., 2016) and distance-constrained interpolations (Chang et al., 2014; Friberg et al., 2016).

Even though many datasets have been developed through deliberately designed statistical models, long-term observations and extensive explanatory variables, there are scientific gaps in many circumstances following this paradigm to develop air pollutant fields. First, these models usually rely on long-term and large-scale station observations for training, especially for those complex time and space resolved models (Feng et al., 2020; Huang et al., 2021). For newly-setup, temporal, or mobile observation networks, there would be limited datasets for training effective data fusion models. Second, most of the previous methods cannot well fuse observations of multiple variables from different monitoring networks. For example, stations in air quality and meteorology observation networks are usually not spatially aligned. The observations in two networks could not be well directly fused in current models. Instead, meteorology reanalysis data were often used as important explanatory variables in previous fusion models (Geng et al., 2015; Ma et al., 2015; Wei et al., 2021). However, in near real-time operational data fusion applications, these reanalysis data would be unavailable or requiring intensive computations. Last but not the least, most of the previous methods that fusing CTM simulations rely on relatively accurate and stable simulation data to achieve good fusion performance (Tong and Mauzerall, 2006). Consistency in CTM parameters, configurations and inputs are also strictly required to guarantee stable fusing performance. Especially in near real-time operational data fusion applications, adjoint models need to be running simultaneously (Friberg et al., 2016) which is costly in computations.

To address these scientific gaps, this study proposed a new data fusion paradigm by designing a deep-learning-based model framework to estimate reanalysis from station observations by learning spatio-temporal correlations from deterministic CTM models. Distinct from the existing data fusion models, the data fusion model does not use any station observations to fit. Instead, the deep learning network was trained with only CTM model simulations to learn their dynamic spatial correlations, which is backed by the CTM's first principals. In the prediction procedure, the fusion/reanalysis data are then generated by the trained model through applying the learned dynamic correlations on real observations. The model framework is fundamentally an alternative of generating chemical reanalysis fields but without rerunning CTMs with data assimilation.

## 2 Data and Methods

### 2.1 CTM Simulations

In this study, the data fusion model was trained to learn spatial correlations of multiple variables from CTM simulations. The simulated PM$_{2.5}$ and other meteorological variables in 2016~2020 were produced using a modeling system that consists of three major components: The meteorology component (WRFv3.4.1) provides meteorological fields, the emission component provides gridded estimates of hourly emissions rates of primary pollutants that matched to model species, and the CTM component (CMAQ v5.0.2 (Byun and Schere, 2006)) solves the governing physical and chemical equations to obtain 3-D pollutant concentrations fields at a horizontal resolution of 12 km. The system was operated at forecasting mode which each day produces CTM simulations for five days ahead. Therefore, corresponding to each day, there are 5 CTM simulations with different forecasting lead time. The CTM simulations of PM$_{2.5}$ concentrations have reasonable performance when evaluated against surface measurements, with root mean square error (RMSE) being 29.28~31.08 μg/m$^3$ and coefficient of determination (R$^2$) being 0.31~0.42 (Figure S1 in the supporting information, SI). The data covered the whole China with 372×426 12-km

by 12-km grid cells. Simulation data in 2016~2019 period were used as the training dataset, while the 2020 simulation data was used for evaluation.

We used the simulated daily mean surface-layer PM$_{2.5}$ concentrations, relative humidity (RH) and wind speed (WS) in the data fusion model. The two meteorological variables are selected because they exhibited stronger correlations with PM$_{2.5}$ concentrations (Figure S2 in SI). Precipitation is found not well correlated with PM$_{2.5}$ concentrations. Boundary layer height (PBL) has relatively strong correlations with PM$_{2.5}$ concentrations, but PBL observation data are not commonly available like other meteorological variables such as RH and WS. Therefore, precipitation and PBL were not included in the model. The air

temperature was not included in the model either because it's highly correlated with surface elevations.

## 2.2 Ground Observations

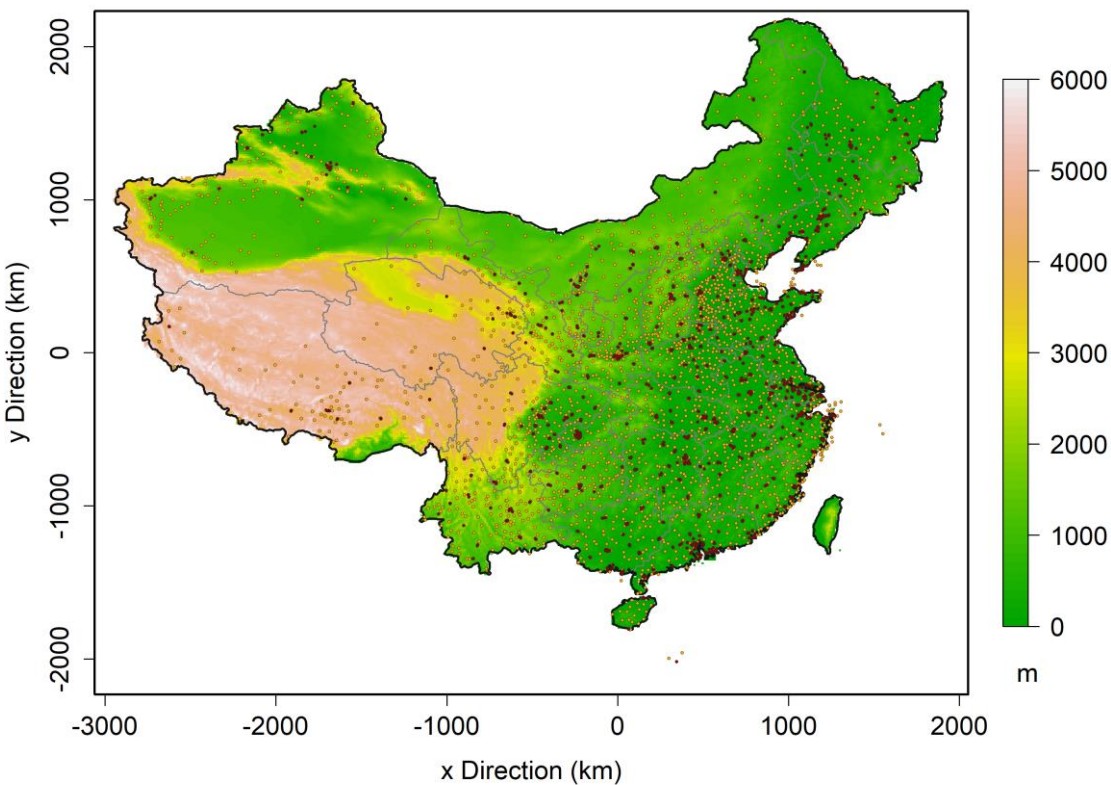

**Figure 1: The map of the study area with elevation in color. Dark red dots represent the national PM$_{2.5}$ monitors and orange dots refer to national meteorological stations.**

PM$_{2.5}$ observations in 2020 were obtained from the China National Environmental Monitoring Center (CNEMC) (http://106.37.208.233:20035/), with the monitoring network as exhibited in Figure 1. Meteorological variables of daily mean RH and WS for the same period at national meteorological observing stations were obtained from the China Meteorology Agency (CMA) network (Figure 1). Geographical variables such as the surface height of Digital Elevation Model (DEM), land use and land cover (LULC) (Zhang et al., 2020) were also used in this study for fusion. From LULC, fraction of urban area is

used to indicate emission strengths. These data variables were resampled to the afore-mentioned CTM simulation grid.

The raw data of both PM$_{2.5}$ and meteorology data were hourly, which were averaged to daily mean if there are more than 18 valid hourly observations in a day at the local time at each monitor. Each of these data items at each station were assigned to a grid that was defined same as used in the CTM simulations. For the sites that co-located in a same grid cell, their averages were used. It should be noted that those grid cells, which do not have valid observations within them, were filled with zeros.

100 In this way, each variable of PM$_{2.5}$, RH and WS will have one gridded observation field in each day.

## 2.3 Deep Learning Data Fusion Framework

The task of obtaining spatially complete air pollutant field from point observations can be regarded as solving a downscaling problem, which means that data values in gap areas among stations need to be optimally estimated from known sparse measurements based on physical or statistical constrains. Most previous studies use statistical methods to relate observations

105 with other supporting variables at stations (Di et al., 2016; Beloconi et al., 2016). In this study, we built a point-to-grid downscaling model by learning from CTM simulations to generate gridded data fusion fields from station observations.

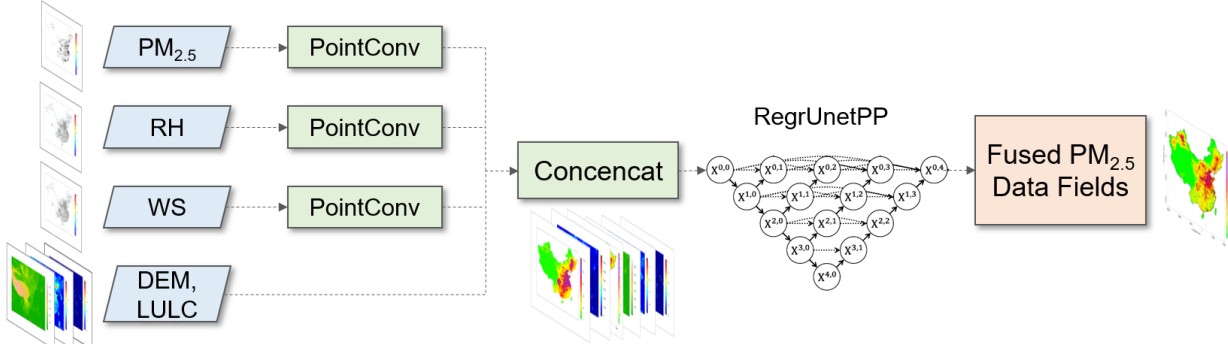

**Figure 2: Data fusion framework using station observations of multiple variables to obtain gridded fields of PM$_{2.5}$.**

A new deep learning model workflow (Figure 2) was designed to fulfill the task of point-to-grid data fusion and downscaling.

110 This deep learning model includes novel point convolutional (PointConv) operations and a backbone fusion module of regression using Unet++ (RegrUnetPP). The PointConv is designed for handling spatially isolated and irregular station observation data to compensate efficacy loss of ordinary convolutions in processing these data. In traditional convolutional operations, the 3×3 filters were often used to calculate moving sum, which would lose effectiveness when it comes to spatially imbalanced station observations. For example, when convolutional filters move to areas without observations, the result will

115 be zero. However, if the convolutional filters work in areas with dense observations, the results will become significantly larger. Therefore, they will generate spatially biased and distorted results (Qi et al., 2018). To solve the problem, we proposed a novel and interpretable PointConv operation to handle isolated station observations of multiple variables. The successive PointConv operation is defined as follows,

$$PointConv_1(x) = \frac{Conv(w_{n1}, x)}{Conv(w_{n1}, x\_one) + e^{-5}} \quad (1)$$

$$PointConv_2(x) = \frac{Conv(w_{n2}, PointConv_1(x) - x)}{Conv(w_{n2}, x\_one) + e^{-5}} \quad (2)$$

$$PC(x) = PointConv_1(x) + PointConv_2(x) \quad (3)$$

Where $w_n$ refers to a convolutional filter with a size $n$. The $Conv(w_n, x)$ in Eq. (1) refers to the ordinary convolution on $x$ with filters $w_n$, which are station observations assigned to pre-defined grid cells. The $x\_one$ was binarized from $x$ by replacing grid cells with valid observation data in $x$ as 1. The PointConv was conducted by mimicking successive analysis procedures

125 as in Eq. (2). The PointConv filter size in the two steps was determined to be 21 and 11 respectively for $n_1$ and $n_2$. In summary, this PointConv operation has the following features and advantages compared to conventional convolutions:

1) The weighted average of isolated data items is calculated rather than weighted sum by only considering valid data,

2) Large-size filters are used to learn and well represent spatial correlations within a large area,

3) Successive PointConv operations are implemented to better reflect spatial variations in local scales,

130 4) Multi-variable observations could be handled separately and simultaneously even if they are from different networks.

The PointConv filters in well-trained models are expected to have larger values in the center area and lower values in the outer

area. With the PointConv module, the spatially complete gridded data set are constructed for three observational variables, denoted as $PC(PM_{2.5})$, $PC(RH)$ and $PC(WS)$. By binding PointConv results of different variables of PM$_{2.5}$, RH and WS with other static supplementary data such as DEM and LULC, input data to data fusion module RegrUnetPP is built. The whole data fusion model could be summarized as in Eq. (4).

$$\hat{y}_{PM_{2.5}} = RegrUnetPP\big(Concencat(PC(PM_{2.5}), PC(RH), PC(WS), DEM, LULC)\big) \quad (4)$$

$$loss = \frac{1}{N}\sum_{i=1}^{N}\big|y_{PM_{2.5},i} - \hat{y}_{PM_{2.5},i}\big| \quad (5)$$

The operation $Concencat$ refers to binding data fields of different data variables into one multiple-channel dataset. The $\hat{y}_{PM_{2.5}}$ refers to the estimated PM$_{2.5}$ concentrations, $y_{PM_{2.5},i}$ refers to the original CTM simulations with $N$ equals to the number of total grid cells.

The fusion module can be any grid-to-grid deep learning model to estimate fused PM$_{2.5}$ concentrations $\hat{y}_{PM_{2.5}}$. Here we used a regression Unet++ model (Zhou et al., 2018) i.e. RegrUnetPP. The RegrUnetPP model was designed as an Encoding-Decoding type network developed from Unet (Ronneberger et al., 2015). Many skip-connection modules (Yamanaka et al., 2017) were added in the RegrUnetPP (Figure 2) to fully explore spatial correlations in different scales while keeping abundant details in output results. RegrUnetPP was constructed by replacing the SoftMax activation layers with the ReLU layers and adopting a mean absolute error (MAE) loss function (Eq.5) instead of the original MaxEntropy function.

## 2.4 Model Training

The model was trained with the 1-day lead CTM simulations of PM$_{2.5}$, RH, and WS, together with geophysical covariates of DEM and LULC. Since four-year data of 2016~2019 has been used to train the model, the whole training dataset has a data shape of $\boldsymbol{R}^{1461\times372\times426\times5}$. Considering that the deep learning model need to learn point-to-grid spatial correlations from CTM simulations, nominal "station" data were constructed by randomly sampling 1500~2500 data points from gridded simulation data separately for each variable at each time, while raw spatially complete PM$_{2.5}$ simulation data were used as the target gridded "truth" data. The sampling data points number 1500~2500 was determined according to the actual air quality monitoring station density in the middle and eastern China. There are around 700 grid cells with air quality monitoring stations in the middle and eastern China within an area of around 4 million square kilometers. Considering the total area of 9.6 million square kilometers in China, the sampling size was set to be random integers in a range of 1500~2500 to ensure sampling point densities are at a similar level as the density of actual monitoring stations. The sampling size was randomly determined for each training batch (i.e., each day), as such the total size of training data points did not vary much among different years.

The spatial correlations of CTM simulations are backed by physical and chemical principles comprehensively represented in the WRF-CMAQ model. The fusion model was trained with the WRF-CMAQ CTM simulations within China from 2016 to 2019 for 20000 iterations with a batch size of 10 when the loss function became stable in about two hours running on a NVIDIA RTX GeForce 2080Ti GPU card. It should be highly noted that the observation data were not involved in the model training procedure at all. In the model prediction procedure, actual station observations will be used as input to generate fused PM$_{2.5}$ concentration fields. We also trained the model with the 5-day lead CTM simulations for the purpose of evaluating impacts from meteorological uncertainties. Note that each of the 1~5-day lead time CTM simulations are driven by different meteorological forecasts, with 1~5 days lead time respectively. The meteorological uncertainties associated with the 5-day lead CTM simulations are usually higher than that with the 1-day lead CTM simulations.

## 2.5 Model Evaluation

In general, the evaluation was conducted for 2020 which is independent from the training data period of 2016~2019. Specifically, the fitted fusion model was evaluated in two aspects. Firstly, its capabilities to predict the fully gridded model

simulations from isolated sampled simulation data items were assessed using the CTM simulation data in 2020. In this aspect, the station-wise CTM PM$_{2.5}$ simulations were constructed by sampling data in those grid cells with CNEMC stations (for PM$_{2.5}$) or CMA stations (for RH and WS) from raw gridded simulations. By feeding these station-wise simulations and supporting variables into the fusion model, spatially completed grided data are obtained. The fused simulation data are then compared against the corresponding raw CTM PM$_{2.5}$ simulations. The comparison was performed in each day, since there are sufficient data items in daily simulations. It should be noted that only those grid cells located in mainland China area were compared. Statistical metrics of coefficient of determinant (R$^2$), root mean square error (RMSE) and normalized mean absolute error (NME) were calculated for performance evaluation.

For the second aspect, data fusion model performance was evaluated with real station observations using two cross validation methods. Specifically, Leave-Stations-Out cross-validation methods (LSCV) and stringent ten-fold Leave-Cities-Out cross-validation (LCCV) were used to evaluate model performance (Lv et al., 2016). In the LCCV method, all cities with PM$_{2.5}$ stations were randomly split into ten groups, while in the LSCV method all stations were randomly split into ten groups. PM$_{2.5}$ observations in one group of stations were used as independent evaluation data, while the data in remaining nine groups were used in data fusion. This process was iteratively performed ten times to ensure all groups of data have been used for evaluation. Considering that the air quality stations are mostly clustered in urban areas, the LCCV method will better reflect the model's performance in predicting PM$_{2.5}$ concentrations in the remote rural areas than the station-based LSCV method. Statistical metrics of R$^2$, RMSE, and NME are also used for statistical measures.

## 3 Results and Discussions

### 3.1 Model Parameters

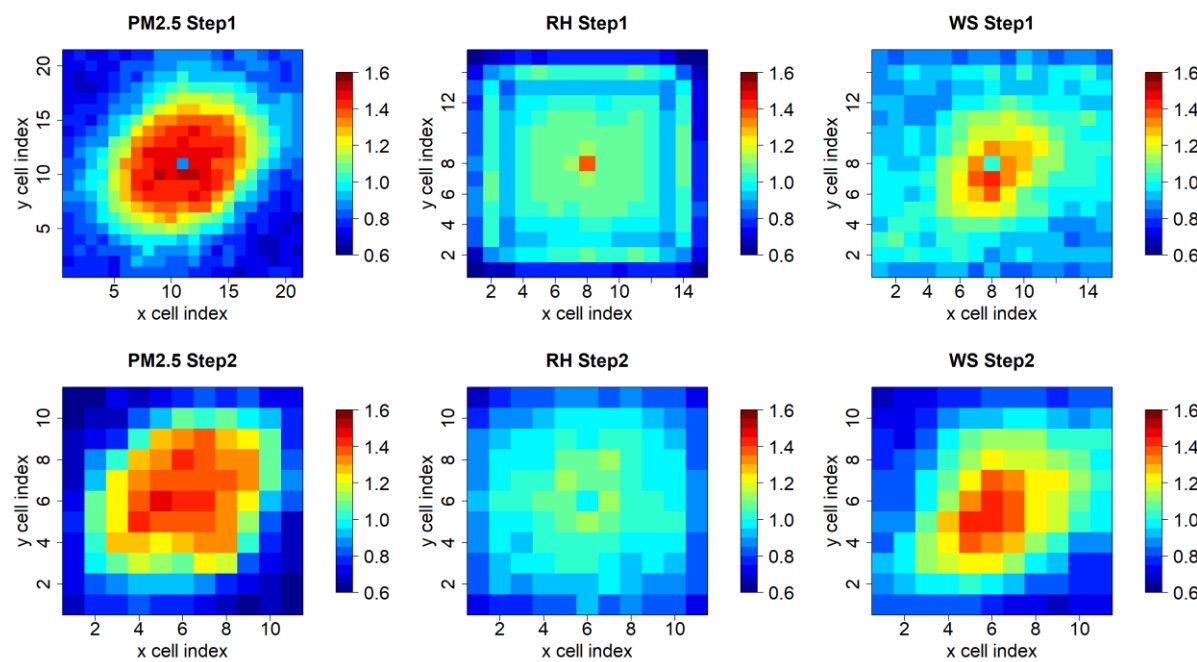

**Figure 3: PointConv filters for PM$_{2.5}$, RH and WS.**

According to its definition, the PointConv was interpretable due to its dedication to implement an interpolation-like process from station observations to remove imbalanced, sparse and clustering distributions of station data items. In fact, the large filters in PointConv resembles to covariance function on distances in common spatial interpolation methods. Larger values in the central area of PointConv filters (Figure 3) indicate spatial correlations are stronger in closely neighboring data items

(Shepard, 1968) than that in long distances. The central values of PointConv filters are respectively around 1.5, 1.1, and 1.4 for PM$_{2.5}$, RH, and WS in both steps. The filters' distribution also revealed that the influencing distance for PM$_{2.5}$ is around 6 grid cells, which was equivalent to 72 kilometers in terms of the 12 km resolution. For RH, the spatial correlations are weak considering that filters were more spatially uniform as exhibited in Figure 3. For WS, it exhibited a stronger locality as indicated by the smaller hot-spot areas with a radius around 4 grid cells (~48 kilometers).

The filters were generally isotropic with slightly larger values in the northeast-southwest direction than in other directions, which could be caused by topographic and climatic patterns in our study area. The slightly anisotropic pattern still exists if wind direction was considered (Figure S3 in SI). Comparing to traditional distance-related interpolation methods such as Kriging and IDW etc. (Friberg et al., 2016), the anisotropy of the filters indicated the PointConv's capability to characterize relatively complex spatial correlations.

## 3.2 Model Performance for Reproducing Simulation Fields

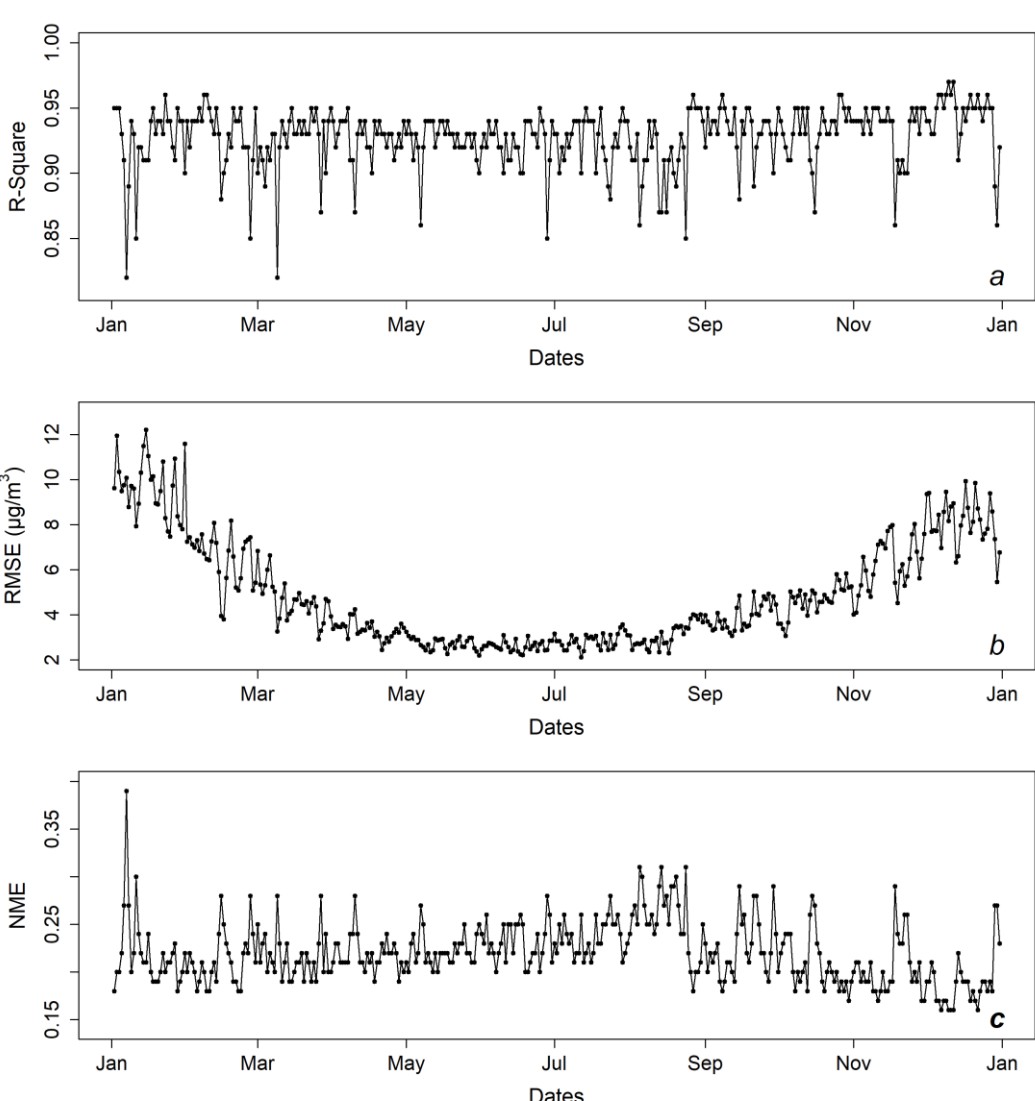

**Figure 4: Daily prediction performance evaluated against the raw PM$_{2.5}$ CTM simulations in 2020: a) R$^2$, b) RMSE and c) NME.**

Our data fusion model has very high accuracies in predicting/reproducing fully gridded PM$_{2.5}$ CTM simulations as exhibited in Figure 4, even though only ~800 PM$_{2.5}$ data points in those grid cells with observation stations were used to estimate data values in the China nation-wide 64488 grid cells. The average of daily R$^2$, RMSE and NME values were respectively 0.94, 4.85 µg/m$^3$ and 0.22 in 2020. The good evaluation metrics indicate the strong capability of the trained deep learning data fusion model reproducing the spatial correlations among multiple variables of the CTM simulations. The fusion model has a stable

performance in terms of $R^2$ and NME values. There are occasional days where $R^2$ values are at low levels of ~0.85. In these
215     days, PM$_{2.5}$ pollution patterns change fast, which were generally less trained compared to those days with more stable patterns.

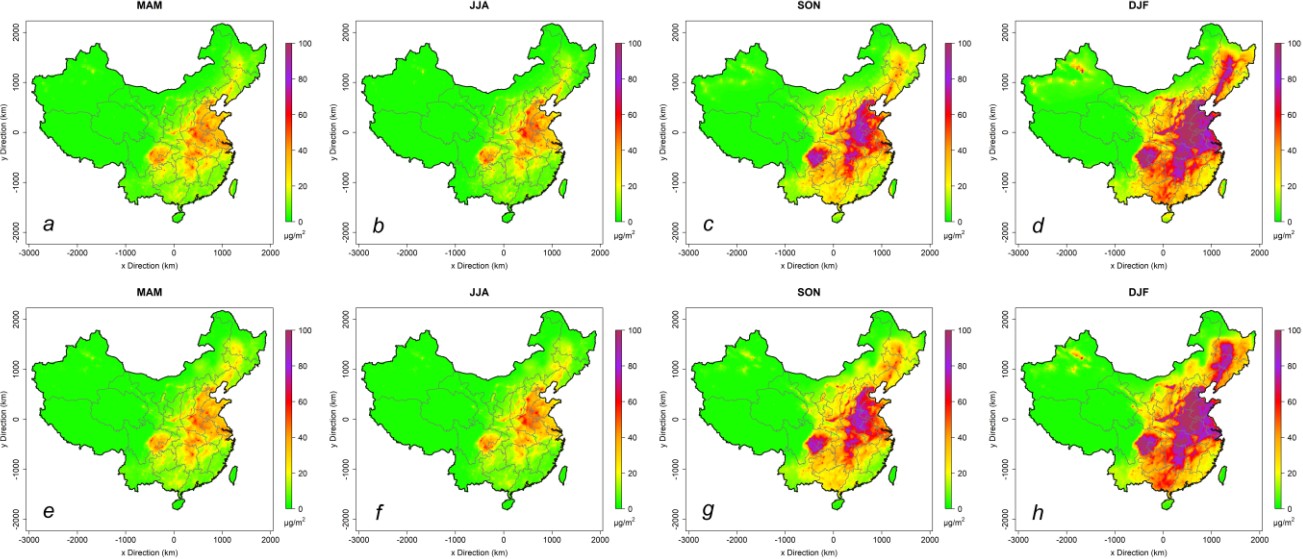

**Figure 5: The seasonal average PM$_{2.5}$ concentrations of the raw CTM simulations (*a* to *d*, first row), and the reproduced simulations using the data fusion model (*e* to *h*, second row).**

By comparing the raw daily average PM$_{2.5}$ CTM simulations and the reproduced PM$_{2.5}$ fields from ~800 data points (Figure
5), it can be concluded that they exhibited high correlations and similarities. The fusion model fully reproduced the raw CTM
simulations in terms of concentration levels, spatial patterns, and fine-scale hot spots, indicting the data fusion model's
capabilities to encode high-level and detailed spatial correlations. By giving the fusion model only a small portion of
simulations that at sparsely scattered locations, it can reproduce the entire whole domain simulation dataset accurately.

### 3.3 Model Performance for Generating Reanalysis Fields

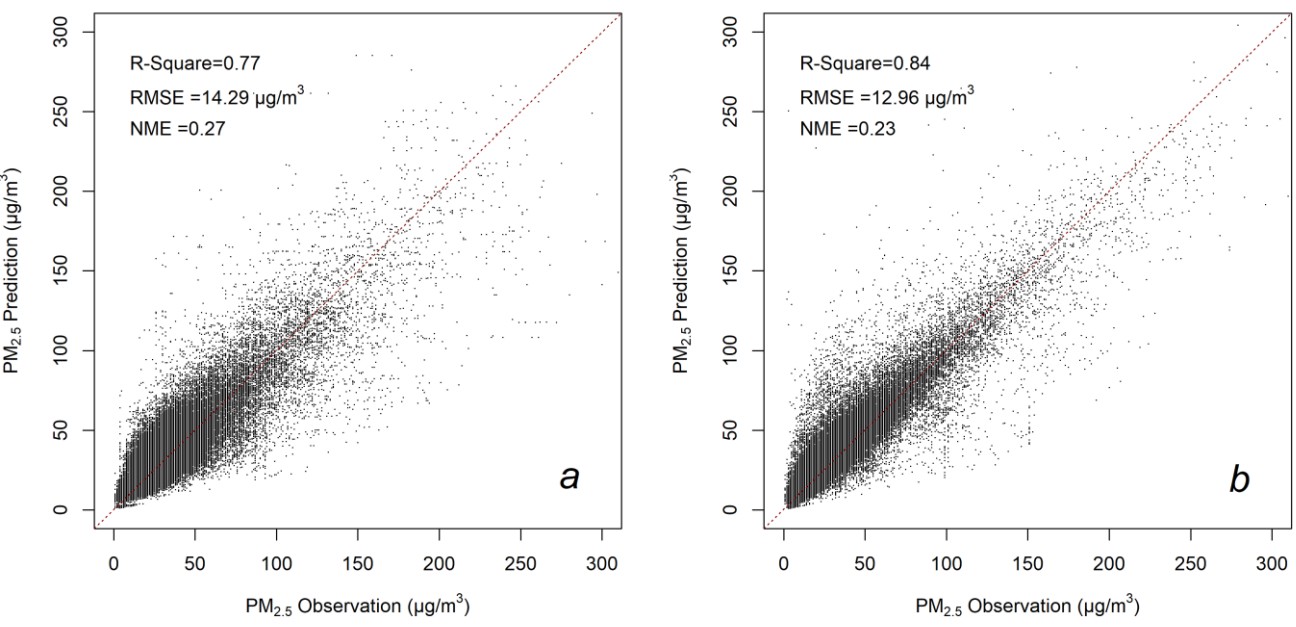

**Figure 6: Scatter plots of predictions versus observations evaluated respectively by the method of a) LCCV and b) LSCV.**

We implemented the data fusion model with the $PM_{2.5}$ and meteorological station observations in 2020 to generate the $PM_{2.5}$ reanalysis fields for evaluation. The evaluation results exhibited good performance with $R^2=0.77$ for the LCCV method and $R^2=0.84$ for the LSCV method (Figure 6), with the RMSE values being respectively 14.29 and 12.96 μg/m³. Considering that most grid cells were located within city urban areas, actual model performance in nation-wide domain should be generally in between the metrics by LCCV and LSCV, which is 0.77~0.84 for $R^2$, 12.96~14.29 μg/m³ for RMSE and 0.23~0.27 for NME. Previous studies tend to underestimate $PM_{2.5}$ concentrations in severe pollution scenarios (Di et al., 2016; Senthilkumar et al., 2019). Our data fusion method predicted high level $PM_{2.5}$ concentrations very well, with NME for $PM_{2.5}$ concentration higher than 150 μg/m³ being small of 0.19 and 0.14 respectively for LCCV and LSCV. It worth noting that there exist increased errors from reproducing CTM simulations ($R^2=0.94$) to generating reanalysis fields ($R^2=0.77~0.84$). The difference of 0.1~0.17 should be mainly attributed to CTM simulation uncertainties of $PM_{2.5}$ spatial correlations compared to actual correlations in observations.

Our model has good performance comparing to the previous studies that used the spatial cross-validation method. For example, Lyu et al. (2019) used an ensemble deep learning model to build relations between CTM simulations and observations of $PM_{2.5}$ in China with a performance of $R^2=0.64$ and RMSE=24.8 μg/m³ using a station-level evaluation method (2019). Xue et al. (2020) fused AOD, CTM simulations and ground observations with a complex multi-stage model and achieved a good performance of $R^2=0.81$ with LSCV method (2017). Xiao et al. (2018) built up an ensemble machine learning model to predict $PM_{2.5}$ at 0.1° resolution with an accuracy of $R^2=0.76$ (2018). Huang et al. (2021) used a multi-stage random-forecast-based model to predict very high-resolution data set and achieved $R^2 = 0.86$ with the LSCV method (2021).

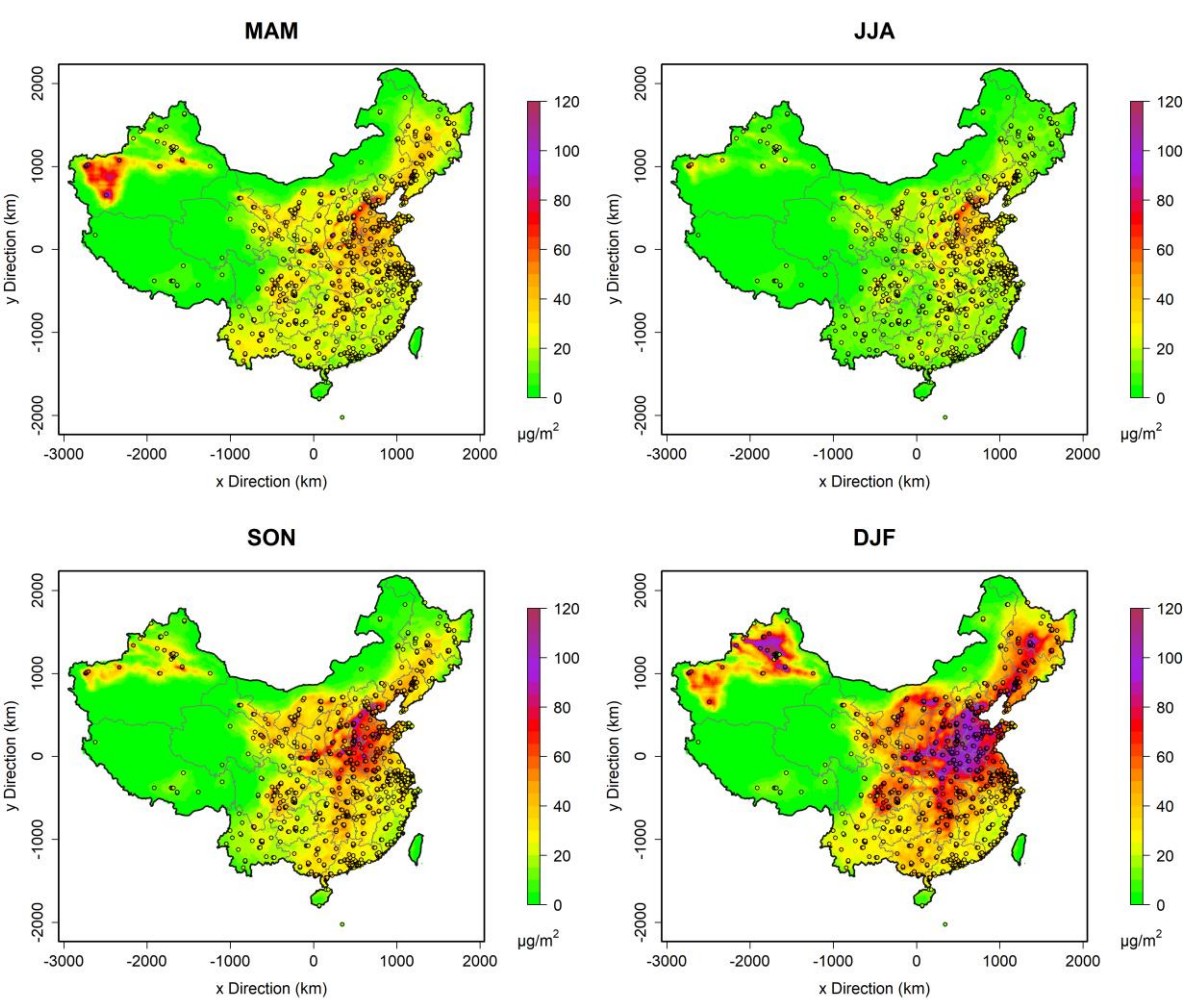

**Figure 7: Reanalysis $PM_{2.5}$ concentration fields in four seasons in 2020. Circles with filled colors represent monitoring sites and corresponding observations.**

Daily PM$_{2.5}$ reanalysis fields for 2020 were obtained with the model framework and station observations. Considering the fused/reanalysis fields are complete in space, the high pollution levels in winter are well revealed in details in the North China Plain (NCP), and in the long-narrow basin areas of Shanxi and Shaanxi provinces (Figure 7). Besides, unlike most previous studies (Huang et al., 2021), PM$_{2.5}$ concentrations in high-altitude clean Tibetan Plateau region are predicted as low values, same as observed. High pollution levels in the middle-west Inner Mongolia area of Hohhot, Baotou, and Yinchuan were also well captured in all four seasons.

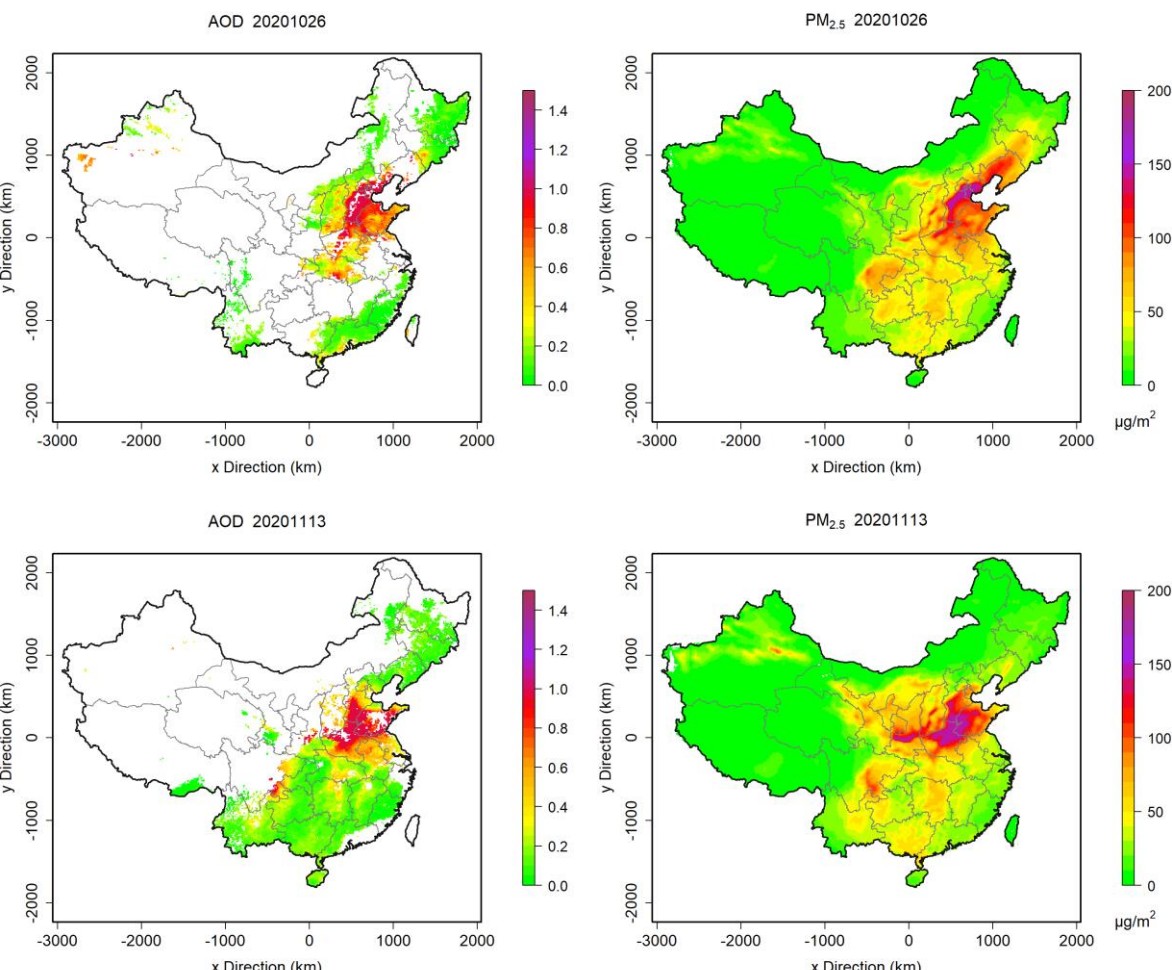

**Figure 8: Comparison between MODIS AOD and PM$_{2.5}$ fusion data on October 26 and November 13 in 2020.**

To further evaluate the spatial distributions of the fused PM$_{2.5}$ fields, we compared them with the MODIS AOD distributions at the days with large AOD spatial coverages (Figure 8 and Figure S4 in SI). Spatial distributions of the fused PM$_{2.5}$ and AOD show high similarities to each other. For example, on October 26, high PM$_{2.5}$ concentrations coincide with high AOD values in NCP, especially in the areas along the east edge of the Taihang Mountains. In detail, PM$_{2.5}$ concentrations and AOD values are both relatively low in the Yimeng Mountains that located in the middle south of Shandong province. The high PM$_{2.5}$ concentrations in the basin area of Shanxi province are also higher than the surrounding area, consistent with that of AOD. On November 13, PM$_{2.5}$ concentrations are extremely high in NCP and high in central China areas of Hubei and Hunan provinces, same as that of AOD values. Besides, in the northeast regions of Yunnan province, both PM$_{2.5}$ and AOD values are relatively high. The spatial coincidence of PM$_{2.5}$ concentrations and AOD values at all levels further validates the accuracy of the fused data.

**4 Discussions**

In this study, $PM_{2.5}$ fields are fused using multiple observational variables from different networks by developing a novel deep learning data fusion model framework. The core of our fusion model is to learn and encode spatial correlations from CTM model simulations to build connections between isolated data points with gridded data fields and among multiple variables. In other words, our method is to "interpolate" the isolated observations to achieve full spatial coverage by using the spatial correlations learned from the CTM simulations. The training is done only with the simulated data. Observational data are not used in the training procedure, but only used in the prediction step. Then the trained fusion model is applied to predict reanalysis/fused data fields from isolated station observations by decoding the learned spatial correlations. As we have demonstrated, the method can accurately reproduce the whole-domain $PM_{2.5}$ concentration fields from only a small number of data points.

In previous studies, all variables need to be spatially paired at stations first to train regression models (Lyu et al., 2019; Xue et al., 2019). To use data from different networks, interpolation and analysis/reanalysis need to be carried out first, which procedure is disconnected from the data fusion model. Here with the successive PointConv modules, it can fuse station data variables from different observation networks, even if they are not spatially aligned at collocations. The PointConv modules were trainable as part of the whole deep learning data fusion model. Without data pairing procedure, the model training and prediction procedure became straightforward that only requires a same spatial grid setting for all input variables.

As we stressed, this model was fitted with model simulation data by learning daily spatial patterns from long-term CTM simulations. It has two benefits. First, the trained deep learning data fusion model can represent and reflect spatial correlations between $PM_{2.5}$ (and any other model species/variables as well) and its supporting variables, by retaining physical and chemical principles in the WRF-CMAQ model. Hence, the method can be readily applied for other CTM simulated species that with measurements available. Second, it doesn't need any observation data sets to train the model. This is quite beneficial for data fusion applications, especially when station networks are newly setup or observations are from mobile or portable sensors. The data fusion models used in previous studies are often requiring long-term observations (Wei et al., 2021; Huang et al., 2021; Xue et al., 2019), which makes it difficult to be reproducibly used in new applications. Conversely, our method is straightforward to use and can be easily examined by inter-comparison with other methods. It provides a pre-trained deep learning model for its application in other studies. To run this model, users only need air quality observations, meteorological observations, and static variables.

It worth noting that high fusion accuracy has been achieved even though the CTM model simulations themselves have relatively low accuracies (Figure S1). CTM model simulation errors usually come from three major sources, which include meteorology uncertainties, emission inventory uncertainties, and imperfect atmospheric physical and chemical parameterizations. However, as we have stated above, the training here is purely to learn the spatial correlations among simulated variables from CTM model simulations, the biases and errors in CTM simulation don't impact the training results. The nearly perfect reproduction of the 2020 simulations has demonstrated this.

To evaluate/separate fusion errors caused by meteorological uncertainties, we trained two data fusion models using respectively the 1-day lead CTM forecasting simulations and the 5-day lead CTM forecasting simulations. The 1-day lead forecasts have different but usually smaller errors than the 5-day lead forecasts (Figure S1). As shown in Figure S5 (see in SI), both trained models have similar performance to produce reanalysis fields. Considering that errors of simulations at different lead time are mainly caused by meteorological inputs, it revealed that CTM errors from meteorology don't have much influences on reanalysis performance.

As for the errors raised by emission uncertainties, we compared the performance of the reanalysis fields in February, March, and April of 2020 against that in other months of 2020. In the three select months, air pollutant emissions in China have dramatically decreased to a very low-level, due to a large-scale national lockdown to prevent the spreading of Covid-19.

Compared to the other months without national lockdowns, emissions inventories used in CTM simulations should have significantly increased uncertainties during these three months. However, the reanalysis performance in the lockdown period does not decrease comparing to that in other months as shown in Figure S6 (see in SI). In fact, input changes of pollutant emissions and meteorological fields within the CTM simulations are allowed and should be encouraged to cover a wider range of emissions and meteorological scenarios to help improve the robustness of the trained model.

However, the uncertainties of physical and chemical parameterizations in CTM could influence the reanalysis performance, as they determine the inherent spatio-temporal correlations among multiple variables. But their impacts on reanalysis performance here are expected to be small as well, provided the configurations in CTM simulations are not changed dramatically, because the nonuniformity of the biases and errors these parametrization uncertainties alone can cause in CTM simulations are expected to be in even less magnitude. We don't recommend using totally different configurations in CTM simulations. Comparatively in other observation-simulation regression methods, both model configurations and meteorology/emission inputs are required to be unchanged and relatively accurate in training data sets (Xue et al., 2017; Hao et al., 2016).

It also worth noting that, the model framework in this study has significant benefit of very high computational efficiency, with computing time for one-time fusion far less than $1s$ running on a consumer GPU card of NVIDIA 2080Ti.

***Data and code availability.*** The CTM simulation data and fused datasets can be accessed by contacting the corresponding authors Baolei Lyu (baoleilv@foxmail.com) and Ran Huang (ranhuang2019@163.com). The land use and land cover data are available at Data Sharing and Service Portal of Chinese Academy of Science (http://data.casearth.cn/en/sdo/detail/5ebe2a9908415d14083a4c24). The source code and a pre-trained model file of the exact version used to produce the results used in this paper is available at https://doi.org/10.5281/zenodo.5152567 on Zenodo (Lyu, 2021). The configuration files for running models of WRF v3.4.1 and CAMQ v5.0.2 are also available at https://doi.org/10.5281/zenodo.5152621 (Hu, 2021).

***Author contributions.*** BL and YH conceived the study. BL developed the model and codes. RH and XW contributed the CTM simulation data. BL and RH collected the observation data. BL analyzed data and wrote the paper with contributions from YH, RH, WW and XW. RH managed the project.

***Competing interests.*** The authors declare that they have no conflict of interest.

***Acknowledgements.*** This research has been in part supported by the National Key R&D Program of China (2018YFC0214000) and the AiMa R&D Project (R#2016-004) of Hangzhou AiMa Technologies. The findings in this research do not necessarily reflect the views of the sponsors.

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
