# Peer review of "Deep-Learning Spatial Principles from Deterministic Chemical Transport Model for Chemical Reanalysis: An Application in China for $PM_{2.5}$"

_Geoscientific Model Development, 2021_

## Author Comment (AC1)

**Authors' Comments on GMD-2021-253**

We would like to thank both reviewers for their thoughtful comments on our study.

**Reviewer 1**

In this paper, the authors developed a deep-learning based method for fusing observational data into simulated air pollution concentration fields. The method requires considerable amount of efforts in preparation, but model execution is fast and the results are favorable, making it suitable for operational air quality forecasting platforms. Overall, this paper is well-structured, fluently written, and the topic fits the scope of this journal.

**Response**: Thanks.

I only have a few comments:

1、Is there any reason why only RH and WS were included as meteorological variables, but other important variables such as precipitation and boundary layer height were not included? Is it because of limitations in computational resources?

**Response**: We have investigated the meteorological variables which could be potentially good predictors by calculating the correlations between each of the common meteorological variables and $PM_{2.5}$ concentrations, all the CTM simulations. We found RH and WS are the most two important meteorological variables for predicting $PM_{2.5}$ concentrations. The precipitations are not well correlated with $PM_{2.5}$ concentrations as exhibited in the following figure. Boundary layer height (PBL) has relatively strong correlations with $PM_{2.5}$, but PBL observation data are not commonly available like other meteorological variables such as RH and WS. Therefore, precipitation and PBL were not included. Besides, air temperature is also not included in the model since it is often highly correlated with surface height of DEM.

We added the following explanation in text: "*The two variables are selected because they exhibited stronger correlations with simulated $PM_{2.5}$ concentrations (Figure S2 in the SI). Even though planetary boundary layer is also relatively well correlated with $PM_{2.5}$, it is not included in the model due to limited availability of observations.*" The following figure was included in the supplementary material as Figure S2.

[Figure]

Figure S2 Boxplots of correlation coefficients between $PM_{2.5}$ concentrations and four select meteorological variables, all simulated by CTM

2、In model training, actual observation data is not used, rather a random sampling of 1500-2500 data points were used. In evaluation, actual observational data were used. It would benefit the readers if the authors could provide more justifications on: 1) Why 1500-2500 data points were chosen, and why the number of data points varies among years; 2) Why random samples of data points were used in training, instead of actual observational data; 3) Will the biases and errors in CTM simulation impact training results?

**Response**:

1) The sampling data points number 1500~2500 was determined according to the actual air quality monitoring station density in the middle and eastern China. There are around 700 grid cells with air quality monitoring stations in the middle and eastern China within an area of around 4 million square kilometers. Considering the total area of 9.6 million square kilometers in China, the sampling size was set to be random integers in a range of 1500~2500 to ensure sampling point densities are at a similar level as the density of actual monitoring stations. The sampling size was randomly determined for each training batch (i.e., each day), as such the total size of training data points did not vary much among different years.

2) The core of our fusion model is to learn spatial correlations from CTM model simulations to build connections between isolated data points with gridded data fields, all simulated variables. Then the trained fusion model is applied to predict analysis/fusion data fields from isolated station observations by adopting the learned spatial correlations among simulated variables. So the training is done only among the simulated variables. Observational data are not used in the

training procedure, but only used in the prediction step. The real novelty of our method is to "interpolate" the isolated observations to achieve full spatial coverage by using the spatial correlations learned from the CTM simulations. The spatial correlations from CTM simulations are considered better than any other available spatial correlation information provided by distance inverse weighting, Kriging, or other means.

3) CTM model simulation errors usually come from three major sources, which include meteorology uncertainties, emission inventory uncertainties, and imperfect atmospheric physical and chemical parameterizations. However, as we have stated above, the training here is purely to learn the spatial correlations among simulated variables from CTM model simulations, so the biases and errors in CTM simulation don't impact on the training results. The nearly perfect performance of evaluation using 2020 simulations has demonstrated this. The trained fusion model predictions (using withheld simulation points) have reproduced the 2020 simulations perfectly at almost every day except for a small number of extreme days. The larger discrepancies on those extreme days are due to smaller pools of data for training, but not because of the biases and errors in the CTM simulations themselves.

However, the quality of the CTM simulations still do affect the performance of the reanalysis results that using observations. This is shown by the worsen performance of the 2020 reanalysis fields compared with the reproduced 2020 simulation fields. The imperfection of the simulated spatial correlations provided by CTM compared to the actual spatial correlations has caused the worsen performance. But it's not the absolute biases and errors in CTM simulations directly impacted on the reanalysis results, but instead the nonuniformity of the biases and errors in CTM simulations over space that twisting the actual gradients between observation spots to non-observations spots is the ultimate confounder. We now have assessed the impacts of CTM simulations' uncertainties on performance of reanalysis fields as follows and found such influences are small.

To evaluate/separate fusion errors caused by meteorological uncertainties, we trained two data fusion models using respectively the 1-day lead CTM forecasting simulations and the 5-day lead CTM forecasting simulations. The 1-day lead forecasts are expected to have different but usually smaller errors than the 5-day lead forecasts. As shown in the following Figure S5, both trained models have similar performance to produce reanalysis fields. The RMSE values of reanalysis fields obtained by using the two trained models are respectively 14.09 and 13.53 $\mu g/m^3$ using the LCCV evaluation method, and 13.04 and 12.17 $\mu g/m^3$ using the LSCV method. This indicated that CTM simulation errors coming from input meteorological fields has little influence on performance of obtained reanalysis fields.

As for the errors raised by emission uncertainties, we compared the performance of the reanalysis fields in February, March, and April of 2020 against that in other months of 2020. In the three select months, air pollutant emissions have been significantly decreased to a very lowlevel, due to a large-scale national lockdown to control the Covid-19 spreading. In other words, comparing to the other months without national lockdowns, emissions inventories used in CTM simulations should have significantly increased uncertainties during these three months. However, the reanalysis performance in the lockdown period does not decrease comparing to that in other months as shown in Figure S6. The RMSE values of reanalysis fields for the two periods are respectively 13.07 and 14.57 $\mu g/m^3$ using the LCCV evaluation method, and 11.76 and 13.33 $\mu g/m^3$ with the LSCV method. This performance comparison revealed that simulation errors in CTM simulations caused by emission inventories won't have significant impact on the performance of the reanalysis.

The uncertainties of physical and chemical parameterizations in CTM could influence the reanalysis performance, as they determine the inherent spatio-temporal correlations among multiple variables. But their impacts on reanalysis performance here are expected to be small as well, provided the configurations in CTM simulations are not changed dramatically, because the nonuniformity of the biases and errors these parametrization uncertainties alone can cause in CTM simulations are expected in even less extent. We don't recommend to use totally different configurations in CTM simulations.

We expanded the related discussions in the manuscript and added supplementary material as follows, "*Considering that the model training process is to fully learn the general spatial correlations among different variables, CTM simulations errors will not influence on training results. In fact, input changes of pollutant emissions and meteorological fields within the CTM simulations are allowed and should be encouraged to cover a wider range of emissions and meteorological scenarios to help improve the robustness of the trained model. To evaluate influences by meteorology input uncertainties on reanalysis results, two deep learning models are trained using simulations in different forecasting lead time. They exhibited same-level accuracies (Figure S5), indicating simulation errors from meteorology have small influences on reanalysis performance. Besides, the reanalysis fields also have similar accuracy levels in lockdown period due to Covid-19 and in non-lockdown period (Figure S6), revealing emission inventory uncertainties will not influence much on model performance either. However, substantial changes of atmospheric physical and chemical principles in CTM models can deteriorate the deep learning model performance because it modifies the simulated correlations between different variables. Comparatively in other observation-simulation regression methods, both model configurations and meteorology/emission inputs are required to be unchanged and relatively accurate in training data sets (Xue et al., 2017; Hao et al., 2015).*".

[Figure]

Figure S5 The evaluation performance for deep learning models trained using the 1-day lead forecasting simulations (panel *a*, *c*) and the 5-day lead forecasting simulations (*b*, *d*) respectively using the LCCV (*a*, *b*) and LSCV (*c*, *d*) methods.

[Figure]

Figure S6 The evaluation performance for deep learning models in the national lockdown period of February to March (panel *a*, *b*) and in remaining periods (*c*, *d*) respectively using LCCV (*a*, *c*) and LSCV (*b*, *d*) methods.

3、In lines 173-174, the authors mentioned that the kernels are generally isotropic but some anisotropic characteristics are evident. What are the expected impacts of such anisotropicity? Does including additional training variables such as wind direction help addressing this anisotropic issue?

**Response**: The anisotropy of the kernel indicated the PointConv's capability to characterize relatively complex spatial correlations comparing to traditional distance-related interpolation methods. In fact, it is not a problem needing to be addressed. Instead, it will help improve model performance. To test wind direction, we used U and V component in the model instead of feeding wind speed into the model. As shown in the following Figure S3, the isotropic effects also exist to reflect spatial correlations of these variables in order to optimally minimize model prediction errors.

The related discussion in the manuscript was added as follows, "*The isotropic pattern still exists if wind direction considered (Figure S3). Comparing to previous only distance-dependent*

*methods (Friberg et al., 2016), the isotropic kernels will have improved representations of varying spatial correlations with directions being also considered.*".

[Figure]

Figure S3 PointConv kernels for PM$_{2.5}$, RH, wind u-component and v-component.

**Reviewer 2**

General Comments

In the manuscript, a new data fusion paradigm is developed to estimate PM2.5 reanalysis fields from station observations by a deep learning framework to learn multi-variable spatial correlations from Chemical Transport Model (CTM) simulations. The model includes an explainable PointConv operation to pre-process isolated observations and a regression grid-to-grid network to reflect correlations among multiple variables. Compared with previous data fusion methods of PM2.5 reanalysis, the proposed fusion framework can fuse multi-variable observations from different monitoring networks (even when they are not spatially aligned at collocations) and the model training does not rely on observations. The deep learning data fusion model framework is novel and can reasonably generate spatio-temporally complete fused fields of PM2.5 using observations at sparse locations. I would recommend publication in Geoscientific Model Development after consideration of the following comments.

**Response**: Thanks.

Specific comments

1. For the proposed fusion framework, why are only the predictions of PM2.5 concentrations, relative humidity (RH) and wind speed (WS) together with the surface height of Digital

Elevation Model (DEM) and land use and land cover (LULC) used to train the deep learning network?

**Response**: The correlations between PM$_{2.5}$ concentration and common meteorological variables have been assessed to determine predictors for training the deep learning network. The correlation coefficients were calculated each day from 2016~2019. The boxplots of coefficients for select important variables exhibited in Figure S2 indicated that relative humidity and wind speed have relatively close correlation higher than other variables. Even though the boundary layer height (PBL) also has relatively strong correlations with PM$_{2.5}$ concentration, PBL observation data are not commonly available like other variables. Therefore, the PBL was not included as a predictor in the model either. Besides, air temperature is also not included in the model since it is often highly correlated with surface height of DEM.

We add the related explanations in the manuscript as follows "*The two variables are selected because they exhibited relatively stronger correlations with PM$_{2.5}$ concentrations (Figure S2 in the SI). Even though planetary boundary layer also well correlated with PM$_{2.5}$, it is not included in the model due to its limited availabilities*". The following figure was included in the supplementary material as Figure S2.

[Figure]

Figure S2 Boxplots of correlation coefficients between PM$_{2.5}$ and four meteorological variables. The correlations are calculated with CTM simulations.

2. Line 247: "This model was fitted with model simulation data by learning daily spatial patterns from long-term CTM simulations." When applying the fusion model, how long period

of CTM simulated data is required at least for the network training to obtain the simulated spatial correlations?

**Response**: The deep learning model was designed to learn the spatial correlations and daily patterns of $PM_{2.5}$ from CTM simulations. While the spatial patterns of $PM_{2.5}$ concentration distributions are mainly determined by daily weather patterns, the daily weather patterns repeat themselves annually with quite small interannual fluctuations. Within the same season, weather patterns often repeat themselves too, but some extreme patterns occur rarely withing a year. We believe, to better cover the spectrum of $PM_{2.5}$ spatial patterns, at least 2-3 years data should be used to train the deep learning model, though the longer the better. We have 5-year CTM data and have used 4-year CTM data for training. We used CTM simulations from an operational forecasting system which each day produces $PM_{2.5}$ simulations for five days ahead. Therefore, corresponding to each day, there are 5 CTM simulations with different forecasting lead time. Note that each of these 5 CTM simulations are driven by different weather forecasts, with 1~5 days lead time respectively. These special datasets we used has increased the robustness of our training results, by increasing the quantity of weather patterns going into training by 4 times.

3. Although, as it is said in lines 258-260, CTM simulation theoretically do not need to be very accurate in the model inputs, an accurate or reasonable spatial correlations (or spatial patterns) simulated by the CTM models is necessary for the model deep-learning. There are very limited information on the CTM simulation data used in the study. Have the simulated PM2.5 spatial patterns been evaluated? How about the performance? Please give some necessary introduction or relevant reference.

**Response**: We used CTM simulations from an operational forecasting system which each day produces $PM_{2.5}$ simulations for five days ahead. Therefore, corresponding to each day, there are 5 CTM simulations with different forecasting lead time. The CTM simulation performance at each lead time was added in the supplementary material as Figure S1. Related descriptions were added in the manuscript as follows, "*The CTM simulations of $PM_{2.5}$ concentrations have reasonable performance when evaluated against surface measurements, with root mean square error (RMSE) being 29.28~31.08 μg/m³ and coefficient of determination ($R^2$) being 0.31~42 (Figure S1 in the SI).*".

[Figure]

Figure S1 The CTM simulation/forecast performance at different lead time. The $R^2$ and RMSE values are calculated at each station for each lead time in 2019.

Technical comments

1. Lines 83-84: "Each of these data items at each were assigned…", the word of "site" or "station" is missed after "at each".

**Response**: Revised.

2. Line 185: "(Figure S2 in the SI)", Figure S2 is not found in the SI. Please check it.

**Response**: The figure numbers in the manuscript and supplementary materials were checked and adjusted.

---

## Author Response (AR1)

**Response to Reviewers' Comments on GMD-2021-253**

We would like to thank both reviewers for their thoughtful comments on our manuscript.

**Reviewer 1**

*In this paper, the authors developed a deep-learning based method for fusing observational data into simulated air pollution concentration fields. The method requires considerable amount of efforts in preparation, but model execution is fast and the results are favorable, making it suitable for operational air quality forecasting platforms. Overall, this paper is well-structured, fluently written, and the topic fits the scope of this journal.*
**Response**: Thanks.

*I only have a few comments:*
*1、Is there any reason why only RH and WS were included as meteorological variables, but other important variables such as precipitation and boundary layer height were not included? Is it because of limitations in computational resources?*
**Response**: We have investigated which specific meteorological variables could be potentially good predictors by calculating the correlations between each of the common meteorological variables and the $PM_{2.5}$ concentrations, all from the CTM simulations. We found RH and WS are the most two important meteorological variables for predicting $PM_{2.5}$ concentrations. Precipitation is found not well correlated with $PM_{2.5}$ concentrations as exhibited in the following Figure S2. Boundary layer height (PBL) has relatively strong correlations with $PM_{2.5}$, but PBL observation data are not commonly available like other meteorological variables such as RH and WS. Therefore, precipitation and PBL were not included in the model. Besides, air temperature is not included in the model either because it is often highly correlated with surface height of DEM.

We added the related explanation in Lines 80~85: "*We used the simulated daily mean surface-layer $PM_{2.5}$ concentrations, relative humidity (RH) and wind speed (WS) in the data fusion model. The two meteorological variables are selected because they exhibited stronger correlations with $PM_{2.5}$ concentrations (Figure S2 in SI). Precipitation is found not well correlated with $PM_{2.5}$ concentrations. Boundary layer height (PBL) has relatively strong correlations with $PM_{2.5}$ concentrations, but PBL observation data are not commonly available like other meteorological variables such as RH and WS. Therefore, precipitation and PBL were not included in the model. The air temperature was not included in the model either because it's highly correlated with surface elevations.*". The following figure was included in the supplementary material as Figure S2.

[Figure]

Figure S2 Boxplots of correlation coefficients between $PM_{2.5}$ concentrations and four select meteorological variables, all simulated by CTM

2、 In model training, actual observation data is not used, rather a random sampling of 1500-2500 data points were used. In evaluation, actual observational data were used. It would benefit the readers if the authors could provide more justifications on: 1) Why 1500-2500 data points were chosen, and why the number of data points varies among years; 2) Why random samples of data points were used in training, instead of actual observational data; 3) Will the biases and errors in CTM simulation impact training results?

**Response**:

1) The sampling data points number 1500~2500 was determined according to the actual air quality monitoring station density in the middle and eastern China. There are around 700 grid cells with air quality monitoring stations in the middle and eastern China within an area of around 4 million square kilometers. Considering the total area of 9.6 million square kilometers in China, the sampling size was set to be random integers in a range of 1500~2500 to ensure sampling point densities are at a similar level as the density of actual monitoring stations. The sampling size was randomly determined for each training batch (i.e., each day), as such the total size of training data points did not vary much among different years.

The related texts were revised in Lines 153~158 as follows, "*The samping data points number 1500~2500 was determined according to the actual air quality monitoring station density in the middle and eastern China. There are around 700 grid cells with air quality monitoring stations in the middle and eastern China within an area of around 4 million square kilometers. Considering the total area of 9.6 million square kilometers in China, the sampling size was set*

*to be random integers in a range of 1500~2500 to ensure sampling point densities are at a similar level as the density of actual monitoring stations. The sampling size was randomly determined for each training batch (i.e., each day), as such the total size of training data points did not vary much among different years.*".

2) The core of our fusion model is to learn spatial correlations from CTM model simulations to build connections between isolated data points with gridded data fields, all among simulated variables. Then the trained fusion model is applied to predict reanalysis/fused data fields from isolated station observations by adopting the learned spatial correlations among simulated variables. So, the training is done only among the simulated variables. Observational data are not used in the training procedure but only used in the prediction step. The real novelty of our method is to "interpolate" the isolated observations to achieve full spatial coverage by using the spatial correlations learned from the CTM simulations. The spatial correlations from CTM simulations are considered better than any other available spatial correlation information provided by distance inverse weighting, Kriging, or other means.

We added Lines 267~275 in Discussions to explain this: "*In this study, $PM_{2.5}$ fields are fused using multiple observational variables from different networks by developing a novel deep learning data fusion model framework. The core of our fusion model is to learn and encode spatial correlations from CTM model simulations to build connections between isolated data points with gridded data fields and among multiple variables. In other words, our method is to "interpolate" the isolated observations to achieve full spatial coverage by using the spatial correlations learned from the CTM simulations. The training is done only with the simulated data. Observational data are not used in the training procedure, but only used in the prediction step. Then the trained fusion model is applied to predict reanalysis/fused data fields from isolated station observations by decoding the learned spatial correlations. As we have demonstrated, the method can accurately reproduce the whole-domain $PM_{2.5}$ concentration fields from only a small number of data points.*"

3) CTM model simulation errors usually come from three major sources, which include meteorology uncertainties, emission inventory uncertainties, and imperfect atmospheric physical and chemical parameterizations. However, as we have stated above, the training here is purely to learn the spatial correlations among simulated variables from CTM model simulations, so the biases and errors in CTM simulation don't impact the training results. The nearly perfect performance of evaluation using 2020 simulations has demonstrated this. The trained fusion model predictions (using withheld simulation points) have reproduced the 2020 simulations perfectly at almost every day except for a small number of extreme days. The larger discrepancies on those extreme days are due to smaller pools of data for training, but not because of the biases and errors in the CTM simulations themselves.

However, the quality of the CTM simulations still does affect the performance of the reanalysis results that using observations. This is shown by the worsening performance of the 2020 reanalysis fields compared with the reproduced 2020 simulation fields. The imperfection of the simulated spatial correlations provided by CTM compared to the actual spatial correlations has caused the worsening performance. But it's not the absolute biases and errors in CTM simulations that directly impacted the reanalysis results, but instead, the nonuniformity of the biases and errors in CTM simulations over space that twists the actual gradients between observation spots to non-observation spots is the ultimate confounder. We now have assessed the impacts of CTM simulations' uncertainties on the performance of reanalysis fields as follows and found such influences are small.

To evaluate/separate fusion errors caused by meteorological uncertainties, we trained two data fusion models using respectively the 1-day lead CTM forecasting simulations and the 5-day lead CTM forecasting simulations. The 1-day lead forecasts are expected to have different but usually smaller errors than the 5-day lead forecasts. As shown in Figure 6 and the following Figure S5, both trained models have similar performance to produce reanalysis fields. The RMSE values of reanalysis fields obtained by using the two trained models are respectively 14.29 and 13.53 μg/m$^3$ using the LCCV evaluation method, and 12.96 and 12.17 μg/m$^3$ using the LSCV method. This indicated that CTM simulation errors coming from input meteorological fields has little influence on the performance of the obtained reanalysis fields.

As for the errors raised by emission uncertainties, we compared the performance of the reanalysis fields in February, March, and April of 2020 against that in other months of 2020. In the three select months, air pollutant emissions have dramatically decreased to a very low-level, due to a large-scale national lockdown to prevent the spreading of Covid-19. Compared to the other months without national lockdowns, emissions inventories used in CTM simulations should have significantly increased uncertainties during these three months. However, the reanalysis performance in the lockdown period does not decrease comparing to that in other months as shown in Figure S6. The RMSE values of reanalysis fields for the two periods are respectively 13.07 and 14.67 μg/m$^3$ using the LCCV evaluation method, and 11.76 and 13.33 μg/m$^3$ with the LSCV method. This performance comparison revealed that simulation errors in CTM simulations caused by emission inventories won't have significant impacts on the performance of the reanalysis.

The uncertainties of physical and chemical parameterizations in CTM could influence the reanalysis performance, as they determine the inherent spatio-temporal correlations among multiple variables. But their impacts on reanalysis performance here are expected to be small as well, provided the configurations in CTM simulations are not changed dramatically, because the nonuniformity of the biases and errors these parametrization uncertainties alone can cause in CTM simulations are expected to be in even less magnitude. We don't recommend using

totally different configurations in CTM simulations.

We expanded the discussions by adding Lines 293~320 as follows, "*It worth noting that high fusion accuracy has been achieved even though the CTM model simulations themselves have relatively low accuracies (Figure S1). CTM model simulation errors usually come from three major sources, which include meteorology uncertainties, emission inventory uncertainties, and imperfect atmospheric physical and chemical parameterizations. However, as we have stated above, the training here is purely to learn the spatial correlations among simulated variables from CTM model simulations, the biases and errors in CTM simulation don't impact the training results. The nearly perfect reproduction of the 2020 simulations has demonstrated this.*

*To evaluate/separate fusion errors caused by meteorological uncertainties, we trained two data fusion models using respectively the 1-day lead CTM forecasting simulations and the 5-day lead CTM forecasting simulations. The 1-day lead forecasts have different but usually smaller errors than the 5-day lead forecasts (Figure S1). As shown in Figure S5 (see in SI), both trained models have similar performance to produce reanalysis fields. Considering that errors of simulations at different lead time are mainly caused by meteorological inputs, it revealed that CTM errors from meteorology don't have much influences on reanalysis performance.*

*As for the errors raised by emission uncertainties, we compared the performance of the reanalysis fields in February, March, and April of 2020 against that in other months of 2020. In the three select months, air pollutant emissions in China have dramatically decreased to a very low-level, due to a large-scale national lockdown to prevent the spreading of Covid-19. Compared to the other months without national lockdowns, emissions inventories used in CTM simulations should have significantly increased uncertainties during these three months. However, the reanalysis performance in the lockdown period does not decrease comparing to that in other months as shown in Figure S6 (see in SI). In fact, input changes of pollutant emissions and meteorological fields within the CTM simulations are allowed and should be encouraged to cover a wider range of emissions and meteorological scenarios to help improve the robustness of the trained model.*

*However, the uncertainties of physical and chemical parameterizations in CTM could influence the reanalysis performance, as they determine the inherent spatio-temporal correlations among multiple variables. But their impacts on reanalysis performance here are expected to be small as well, provided the configurations in CTM simulations are not changed dramatically, because the nonuniformity of the biases and errors these parametrization uncertainties alone can cause in CTM simulations are expected to be in even less magnitude. We don't recommend using totally different configurations in CTM simulations. Comparatively in other observation-simulation regression methods, both model configurations and meteorology/emission inputs are required to be unchanged and relatively accurate in training data sets (Xue et al., 2017; Hao et al., 2016).*".

[Figure]

Figure S5 Performance evaluation of the fused PM$_{2.5}$ fields in 2020 using the model trained with the 5-day lead CTM simulations respectively using the LCCV (*a*) and LSCV (*b*) methods.

[Figure]

Figure S6 Performance evaluation of the fused PM$_{2.5}$ fields in the national lockdown period of

February to April in 2020 (panel *a*, *b*) and in the remaining periods (*c*, *d*) respectively using the LCCV (*a*, *c*) and LSCV (*b*, *d*) methods.

3、In lines 173-174, the authors mentioned that the kernels are generally isotropic but some anisotropic characteristics are evident. What are the expected impacts of such anisotropicity? Does including additional training variables such as wind direction help addressing this anisotropic issue?

**Response**: The anisotropy of the kernel (we changed the term "kernel" to "filter") indicated the PointConv's capability to characterize relatively complex spatial correlations compared to traditional distance-related interpolation methods. In fact, it is not a problem needing to be addressed. Instead, it will help improve model performance. To test wind direction, we used U and V components in the model instead of feeding wind speed into the model. As shown in the following Figure S3, the anisotropic effects also exist to reflect spatial correlations of these variables in order to optimally minimize model prediction errors.

The related discussion in the manuscript was added in Lines 202~205 as follows, "*The slightly anisotropic pattern still exists if wind direction was considered (Figure S3 in SI). Comparing to traditional distance-related interpolation methods such as Kriging and IDW etc. (Friberg et al., 2016), the anisotropy of the filters indicated the PointConv's capability to characterize relatively complex spatial correlations.*".

[Figure]

Figure S3 PointConv filters for PM$_{2.5}$, RH, wind u-component and v-component.

**Reviewer 2**

In the manuscript, a new data fusion paradigm is developed to estimate PM2.5 reanalysis fields from station observations by a deep learning framework to learn multi-variable spatial

correlations from Chemical Transport Model (CTM) simulations. The model includes an explainable PointConv operation to pre-process isolated observations and a regression grid-to-grid network to reflect correlations among multiple variables. Compared with previous data fusion methods of PM2.5 reanalysis, the proposed fusion framework can fuse multi-variable observations from different monitoring networks (even when they are not spatially aligned at collocations) and the model training does not rely on observations. The deep learning data fusion model framework is novel and can reasonably generate spatio-temporally complete fused fields of PM2.5 using observations at sparse locations. I would recommend publication in Geoscientific Model Development after consideration of the following comments.

**Response**: Thanks.

Specific comments

1. For the proposed fusion framework, why are only the predictions of PM2.5 concentrations, relative humidity (RH) and wind speed (WS) together with the surface height of Digital Elevation Model (DEM) and land use and land cover (LULC) used to train the deep learning network?

**Response**: The correlations between $PM_{2.5}$ concentration and common meteorological variables have been assessed to determine predictors for training the deep learning network. The correlation coefficients were calculated each day from 2016~2019. The boxplots of coefficients for select important variables exhibited in Figure S2 indicated that relative humidity and wind speed have relatively close correlations higher than other variables. Even though the boundary layer height (PBL) also has relatively strong correlations with $PM_{2.5}$ concentration, PBL observation data are not commonly available like other variables. Therefore, the PBL was not included as a predictor in the model. Besides, air temperature is also not included in the model either since it is often highly correlated with surface height of DEM.

We add the related explanations in the manuscript in Lines 80~85 as follows, "*We used the simulated daily mean surface-layer $PM_{2.5}$ concentrations, relative humidity (RH) and wind speed (WS) in the data fusion model. The two meteorological variables are selected because they exhibited stronger correlations with $PM_{2.5}$ concentrations (Figure S2 in SI). Precipitation is found not well correlated with $PM_{2.5}$ concentrations. Boundary layer height (PBL) has relatively strong correlations with $PM_{2.5}$ concentrations, but PBL observation data are not commonly available like other meteorological variables such as RH and WS. Therefore, precipitation and PBL were not included in the model. The air temperature was not included in the model either because it's highly correlated with surface elevations.*". The following figure was included in the supplementary material as Figure S2.

[Figure]

Figure S1 Boxplots of correlation coefficients between $PM_{2.5}$ concentrations and four select meteorological variables, all simulated by CTM.

2. Line 247: "This model was fitted with model simulation data by learning daily spatial patterns from long-term CTM simulations." When applying the fusion model, how long period of CTM simulated data is required at least for the network training to obtain the simulated spatial correlations?

**Response**: The deep learning model was designed to learn the spatial correlations and daily patterns of $PM_{2.5}$ from CTM simulations. While the spatial patterns of $PM_{2.5}$ concentration distributions are mainly determined by daily weather patterns, the daily weather patterns repeat themselves annually with quite small interannual fluctuations. Within the same season, weather patterns often repeat themselves too, but some extreme patterns occur rarely within a year. We believe, to better cover the spectrum of $PM_{2.5}$ spatial patterns, at least 2-3 years' data should be used to train the deep learning model, though the longer the better. We have 5-year CTM data and have used 4-year CTM data for training. We used CTM simulations from an operational forecasting system which each day produces $PM_{2.5}$ simulations for five days ahead. Therefore, corresponding to each day, there are 5 CTM simulations with different forecasting lead time. Note that each of these 5 CTM simulations is driven by different weather forecasts, with 1~5 days lead time respectively.

3. Although, as it is said in lines 258-260, CTM simulation theoretically do not need to be very accurate in the model inputs, an accurate or reasonable spatial correlations (or spatial patterns) simulated by the CTM models is necessary for the model deep-learning. There are very limited information on the CTM simulation data used in the study. Have the simulated PM2.5 spatial

patterns been evaluated? How about the performance? Please give some necessary introduction or relevant reference.

**Response**: We used CTM simulations from an operational forecasting system which each day produces $PM_{2.5}$ simulations for five days ahead. Therefore, corresponding to each day, there are 5 CTM simulations with different forecasting lead time. The CTM simulation performance at each lead time was added in the supplementary material as Figure S1.

Related descriptions were added in the manuscript in Lines 73~77 as follows, "*The system was operated at forecasting mode which each day produces CTM simulations for five days ahead. Therefore, corresponding to each day, there are 5 CTM simulations with different forecasting lead time. The CTM simulations of $PM_{2.5}$ concentrations have reasonable performance when evaluated against surface measurements, with root mean square error (RMSE) being 29.28~31.08 $\mu g/m^3$ and coefficient of determination ($R^2$) being 0.31~0.42 (Figure S1 in the supporting information, SI).*".

[Figure]

Figure S2 The performance measures as $R^2$ and RMSE of CTM simulations of $PM_{2.5}$ concentration against station measurements at different forecasting lead time in 2019.

Technical comments

1. Lines 83-84: "Each of these data items at each were assigned…", the word of "site" or "station" is missed after "at each".

**Response**: Corrected.

2. Line 185: "(Figure S2 in the SI)", Figure S2 is not found in the SI. Please check it.

**Response**: The figure numberings in the manuscript and supplementary material were checked and adjusted.